# The Double-Sided Information Bottleneck Function [note 1]

**DOI:** 10.3390/e24091321

**Published:** 2022-09-19

**Authors:** Michael Dikshtein, Or Ordentlich, Shlomo Shamai (Shitz)

**Affiliations:** 1Department of Electrical and Computer Engineering, Technion, Haifa 3200003, Israel; 2School of Computer Science and Engineering, The Hebrew University of Jerusalem, Jerusalem 9190401, Israel

**Keywords:** information bottleneck, lossy compression, remote source coding, biclustering

## Abstract

A double-sided variant of the information bottleneck method is considered. Let (X,Y) be a bivariate source characterized by a joint pmf $P_{XY}$. The problem is to find two independent channels $P_{U|X}$ and $P_{V|Y}$ (setting the Markovian structure U→X→Y→V), that maximize I(U;V) subject to constraints on the relevant mutual information expressions: I(U;X) and I(V;Y). For jointly Gaussian X and Y, we show that Gaussian channels are optimal in the low-SNR regime but not for general SNR. Similarly, it is shown that for a doubly symmetric binary source, binary symmetric channels are optimal when the correlation is low and are suboptimal for high correlations. We conjecture that Z and S channels are optimal when the correlation is 1 (i.e., X=Y) and provide supporting numerical evidence. Furthermore, we present a Blahut–Arimoto type alternating maximization algorithm and demonstrate its performance for a representative setting. This problem is closely related to the domain of biclustering.

## 1. Introduction

The *information bottleneck* (IB) method [1] plays a central role in advanced lossy source compression. The analysis of classical source coding algorithms is mainly approached via the rate-distortion theory, where a fidelity measure must be defined. However, specifying an appropriate distortion measure in many real-world applications is challenging and sometimes infeasible. The IB framework introduces an essentially different concept, where another variable is provided, which carries the relevant information in the data to be compressed. The quality of the reconstructed sequence is measured via the mutual information metric between the reconstructed data and the relevance variables. Thus, the IB method provides a universal fidelity measure.

In this work, we extend and generalize the IB method by imposing an additional bottleneck constraint on the relevant variable and considering noisy observation of the source. In particular, let (X,Y) be a bivariate source characterized by a fixed joint probability law $P_{XY}$ and consider all Markov chains U→X→Y→V. The Double-Sided Information Bottleneck (DSIB) function is defined as [2]:(1)$$R_{P_{XY}}\left(C_{u},C_{v}\right)≜\max I\left(U;V\right),$$
where the maximization is over all $P_{U|X}$ and $P_{V|Y}$ satisfying $I\left(U;X\right)\leq C_{u}$ and $I\left(V;Y\right)\leq C_{v}$. This problem is illustrated in Figure 1. In our study, we aim to determine the maximum value and the achieving conditional distributions $\left(P_{U|X},P_{V|Y}\right)$ (test channels) of (Equation 1) for various fixed sources $P_{XY}$ and constraints $C_{u}$ and $C_{v}$.

The problem we consider originates from the domain of clustering. Clustering is applied to organize similar entities in unsupervised learning [3]. It has numerous practical applications in data science, such as: joint word-document clustering, gene expression [4], and pattern recognition. The data in those applications are arranged as a contingency table. Usually, clustering is performed on one dimension of the table, but sometimes it is helpful to apply clustering on both dimensions of the contingency table [5], for example, when there is a strong correlation between the rows and the columns of the table or when high-dimensional sparse structures are handled. The input and output of a typical biclustering algorithm are illustrated in Figure 2. Consider an S×T data matrix $\left(a_{st}\right)$. Find partitions $B_{k}\subseteq \left\{1,\cdots ,S\right\}$ and $C_{l}\subseteq \left\{1,\cdots ,T\right\}$, k=1,⋯,K, l=1,⋯,L such that all elements of the “biclusters” [6] ${\left(a_{st}\right)}_{s\in B_{k},t\in C_{l}}$ are homogeneous. The measure of homogeneity depends on the application.

This problem can also be motivated by a remote source coding setting. Consider a latent random variable W, which satisfies U←X←W→Y→V and represents a source of information. We have two users that observe noisy versions of W, i.e., X and Y. Those users try to compress the observed noisy data so that their reconstructed versions, U and V, will be comparable under the maximum mutual information metric. The problem we consider also bears practical applications. Imagine a distributed sensor network where the different edges measure a noisy version of a particular signal but are not allowed to communicate with each other. Each of the nodes performs compression of the received signal. Under the DSIB framework, we can find the optimal compression schemes that preserve the reconstructed symbols’ proximity subject to the mutual information measure.

Dhillon et al. [7] initiated an information-theoretic approach to biclustering. They have regarded the normalized non-negative contingency table as a joint probability distribution matrix of two random variables. Mutual information was proposed as a measure for optimal co-clustering. An optimization algorithm was presented that intertwines both row and column clustering at all stages. Distributed clustering from a proper information-theoretic perspective was first explicitly considered by Pichler et al. [2]. Consider the model illustrated in Figure 3. A bivariate memory-less source with joint law $P_{XY}$ generates *n* i.i.d. copies $\left(X^{n},Y^{n}\right)$ of (X,Y). Each sequence is observed at two different encoders, and each encoder generates a description of the observed sequence, $f_{n}\left(X^{n}\right)$ and $g_{n}\left(Y^{n}\right)$. The objective is to construct the mappings $f_{n}$ and $g_{n}$ such that the normalized mutual information between the descriptions would be maximal while the description coding has bounded rate constraints. Single-letter inner and outer bounds for a general $P_{XY}$ were derived. An example of a *doubly symmetric binary source* (DSBS) was given, and several converse results were established. Furthermore, connections were made to the standard IB [1] and the *multiple description* CEO problems [8]. In addition, the equivalence of information-theoretic biclustering problem to hypothesis testing against independence with multiterminal data compression and a pattern recognition problem was established in [9,10], respectively.

The DSIB problem addressed in our paper is, in fact, a single-letter version of the *distributed clustering* setup [2]. The inner bound in [2] coincides with our problem definition. Moreover, if the Markov condition U→X→Y→Z is imposed on the multi-letter variant, then those problems are equivalent. A similar setting, but with a maximal correlation criterion between the reconstructed random variables, has been considered in [11,12]. Furthermore, it is sometimes the case that the optimal biclustering problem is more straightforward to solve than its standard, single-sided, clustering counterpart. For example, the Courtade–Kumar conjecture [13] for the standard single-sided clustering setting was ultimately proven for the biclustering setting [14]. A particular case, where (X,Y) are drawn from DSBS distribution and the mappings $f_{n}$ and $g_{n}$ are restricted to be Boolean functions, was addressed in [14]. The bound $I\left(f_{n}\left(X^{n}\right);g_{n}\left(Y^{n}\right)\right)\leq I\left(X;Y\right)$ was established, which is tight if and only if $f_{n}$ and $g_{n}$ are dictator functions.

### 1.1. Related Work

Our work extends the celebrated standard (single-sided) IB (SSIB) method introduced by Tishby et al. [1]. Indeed, consider the problem illustrated in Figure 4. This single-sided counterpart of our work is essentially a remote source coding problem [15,16,17], choosing the distortion measure as the logarithmic loss. The random variable U represents the noisy version (X) of the source (Y) with a constrained number of bits (I(U;X)≤C), and the goal is to maximize the relevant information in U regarding Y (measured by the mutual information between Y and U). In the standard IB setup, I(U;X) is referred to as the complexity of U, and I(Y;U) is referred to as the relevance of U.

For the particular case where (U,X,Y) are discrete, an optimal $P_{U|X}$ can be found by iteratively solving a set of self-consistent equations. A generalized Blahut–Arimoto algorithm [18,19,20,21] was proposed to solve those equations. The optimal test-channel $P_{U|X}$ was characterized using a variation principle in [1]. A particular case of deterministic mappings from X to U was considered in [22], and algorithms that find those mappings were described.

Several representing scenarios have been considered for the SSIB problem. The setting where the pair (X,Y) is a *doubly symmetric binary source* (DSBS) with transition probability *p* was addressed from various perspectives in [17,23,24]. Utilizing Mrs. Gerber’s Lemma (MGL) [25], one can show that the optimal test-channel for the DSBS setting is a BSC. The case where (X,Y) are jointly multivariate Gaussians in the SSIB framework was first considered in [26]. It was shown that the optimal distribution of (U,X,Y) is also jointly Gaussian. The optimality of the Gaussian test channel can be proven using EPI [27], or exploiting I-MMSE and Single Crossing Property [28]. Moreover, the proof can be easily extended to jointly Gaussian random vectors (X,Y) under the I-MMSE framework [29].

In a more general scenario where X=Y+Z and only Z is fixed to be Gaussian, it was shown that discrete signaling with deterministic quantizers as test-channel sometimes outperforms Gaussian $P_{X}$ [30]. This exciting observation leads to a conjecture that discrete inputs are optimal for this general setting and may have a connection to the input amplitude constrained AWGN channels where it was already established that discrete input distributions are optimal [31,32,33]. One reason for the optimality of discrete distributions stems from the observation that constraining the compression rate limits the usable input amplitude. However, as far as we know, it remains an open problem.

There are various related problems considered in the literature that are equivalent to the SSIB; namely, they share a similar single-letter optimization problem. In the *conditional entropy bound* (CEB) function, studied in [17], given a fixed bivariate source (X,Y) and an equality constraint on the conditional entropy of X given U, the goal is to minimize the conditional entropy of Y given U over the set of U such that U→X→Y constitute a Markov chain. One can show that CEB is equivalent to SSIB. The *common reconstruction* CR setting [34] is a source coding with a side-information problem, also known as Wyner–Ziv coding, as depicted in Figure 5; with an additional constraint, the encoder can reconstruct the same sequence as the decoder. Additional assumption of log-loss fidelity results in a single-letter rate-distortion region equivalent to the SSIB. In the problem of *information combining* (IC) [23,35], motivated by message combining in LDPC decoders, a source of information, $P_{Y}$, is observed through two test-channels $P_{X|Y}$ and $P_{Z|Y}$. The IC framework aims to design those channels in two extreme approaches. For the first, IC asks what those channels should be to make the output pair (X,Z) maximally informative regarding Y. On the contrary, IC also considers how to design $P_{X|Y}$ and $P_{Z|Y}$ to minimize the information in (X,Z) regarding Y. The problem of minimizing IC can be shown to be equivalent to the SSIB. In fact, if (X,Y) is a DSBS, then by [23], $P_{Z|Y}$ is a *binary symmetric channel* (BSC), recovering similar results from (Section IV.A of [17]).

The IB method has been extended to various network topologies. A multilayer extension of the IB method is depicted in Figure 6. This model was first considered in [36]. A multivariate source $\left(X,Y_{1},\cdots ,Y_{L}\right)$ generates a sequence of *n* i.i.d. copies $\left(X^{n},Y_{1}^{n},\cdots ,Y_{L}^{n}\right)$. The receiver has access only to the sequence $X^{n}$ while $\left(Y_{1}^{n},\cdots ,Y_{L}^{n}\right)$ are hidden. The decoder performs a consecutive *L*-stage compression of the observed sequence. The representation at step *k* must be maximally informative about the respective hidden sequence $Y_{k}$, k∈{1,2,⋯,L}. This setup is highly motivated by the structure of deep neural networks. Specific results were established for the binary and Gaussian sources.

The model depicted in Figure 7 represents a multiterminal extension of the standard IB. A set of receivers observe noisy versions $\left(X_{1},X_{2},\cdots ,X_{K}\right)$ of some source of information Y. The channel outputs $\left(X_{1},X_{2},\cdots ,X_{K}\right)$ are conditionally independent given Y. The receivers are connected to the central processing unit through noiseless but limited-capacity backhaul links. The central processor aims to attain a good prediction $\hat{Y}$ of the source Y based on compressed representations of the noisy version of Y obtained from the receivers. The quality of prediction is measured via the mutual information merit between Y and $\hat{Y}$. The Distributive IB setting is essentially a CEO source coding problem under logarithmic loss (log-loss) distortion measure [37]. The case where $\left(X,Y_{1},\cdots ,Y_{K}\right)$ are jointly Gaussian random variables was addressed in [20], and a Blahut–Arimoto-type algorithm was proposed. An optimized algorithm to design quantizers was proposed in [38].

A cooperative multiterminal extension of the IB method was proposed in [39]. Let $\left(X_{1}^{n},X_{2}^{n},Y^{n}\right)$ be *n* i.i.d. copies of the multivariate source $\left(X_{1},X_{2},Y\right)$. The sequences $X_{1}^{n}$ and $X_{2}^{n}$ are observed at encoders 1 and 2, respectively. Each encoder sends a representation of the observed sequence through a noiseless yet rate-limited link to the other encoder and the mutual decoder. The decoder attempts to reconstruct the latent representation sequence $Y^{n}$ based on the received descriptions. As shown in Figure 8, this setup differs from the CEO setup [40] since the encoders can cooperate during the transmission. The set of all feasible rates of complexity and relevance were characterized, and specific regions for the binary and Gaussian sources were established. There are many additional variations of multi-user IB in the literature [20,26,35,36,37,39,40,41,42,43,44].

The IB problem connects to many timely aspects, such as *capital investment* [43], *distributed learning* [45], *deep learning* [46,47,48,49,50,51,52], and *convolutional neural networks* [53,54]. Moreover, it has been recently shown that the IB method can be used to reduce the data transfer rate and computational complexity in 5G LDPC decoders [55,56]. The IB method has also been connected with constructing good polar codes [57]. Due to the exponential output-alphabet growth of polarized channels, it becomes demanding to compute their capacities to identify the location of “frozen bits". Quantization is employed in order to reduce the computation complexity. The quality of the quantization scheme is assessed via mutual information preservation. It can be shown that the corresponding IB problem upper bounds the quantization technique. Quantization algorithms based upon the IB method were considered in [58,59,60]. Furthermore, a relationship between the KL means algorithm and the IB method has been discovered in [61].

A recent comprehensive tutorial on the IB method and related problems is given in [24]. Applications of IB problem in *machine learning* are detailed in [26,45,46,47,51,52,62].

### 1.2. Notations

Throughout the paper, random variables are denoted using a sans-serif font, e.g., X, their realizations are denoted by the respective lower-case letters, e.g., *x*, and their alphabets are denoted by the respective calligraphic letters, e.g., X. Let $X^{n}$ stand for the set of all *n*-tuples of elements from X. An element from $X^{n}$ is denoted by $x^{n}=\left(x_{1},x_{2},\cdots ,x_{n}\right)$ and substrings are denoted by $x_{i}^{j}=\left(x_{i},x_{i+1},\cdots ,x_{j}\right)$. The cardinality of a finite set, say X, is denoted by |X|. The probability mass function (pmf) of X, the joint pmf of X and Y, and the conditional pmf of X given Y are denoted by $P_{X}$, $P_{XY}$, and $P_{X|Y}$, respectively. The expectation of X is denoted by $E_{}\left[X\right]$. The probability of an event E is denoted as $P\left(E\right)$.

Let X and Y be an *n*-ary and *m*-ary random variables, respectively. The marginal probability vector is denoted by a lowercase boldface letter, i.e.,
(2)$$q≜{\left(P\left(X=1\right),P\left(X=2\right),\cdots ,P\left(X=n\right)\right)}^{T}.$$

The probability vector of an *n*-ary uniform random variable is denoted by $u_{n}$. We denote by *T* the transition matrix from X to Y, i.e.,
(3)$$T_{ij}≜P\left(Y=i|X=j\right),\;1\leq i\leq m,1\leq j\leq n.$$

The entropy of *n*-ary probability vector q is given by h(q), where
(4)$$h\left(q\right)≜-\sum_{i=1}^{n}q_{i}\log q_{i}.$$

Throughout this paper all logarithms are taken to base 2 unless stated otherwise. We denote the ones complemented with a bar, i.e., $\bar{x}=1-x$. The binary convolution of x,y∈[0,1] is defined as $x*y≜x\bar{y}+\bar{x}y$. The binary entropy function is defined by $h_{b}\left(p\right):\left[0,1\right]\to \left[0,1\right]$, i.e., $h_{b}\left(p\right)≜-p\log p-\bar{p}\log\bar{p}$, and $h_{b}^{-1}\left(\;\cdot \;\right)$ its inverse, restricted to [0,1/2].

Let X and Y be a pair of random variables with joint pmf $P_{XY}$ and marginal pmfs $P_{X}=q_{x}$ and $P_{Y}=q_{y}$. Furthermore, let *T* ($\bar{T}$) be the transition matrix from X (Y) to Y (X). The mutual information between X and Y is defined as:(5)$$I\left(X;Y\right)=I\left(q_{x},T\right)=I\left(q_{y},\bar{T}\right)=\sum_{x\in X}\sum_{y\in Y}P_{XY}\left(x,y\right)\log\frac{P_{XY}\left(x,y\right)}{P_{X}\left(x\right)P_{Y}\left(y\right)}.$$

### 1.3. Paper Outline

Section 2 gives a proper definition of the DSIB optimization problem, mentions various results directly related to this work, and provides some general preliminary results. The spotlight of Section 3 is on the binary (X,Y), where we derive bounds on the respective DSIB function and show a complete characterization for extreme scenarios. The jointly Gaussian (X,Y) is considered in Section 4, where an elegant representation of an objective function is presented, and complete characterization in the low-SNR regime is established. A Blahut–Arimoto-type alternating maximization algorithm will be presented in Section 5. Representative numerical evaluation of the bounds and the proposed algorithm will be provided in Section 6. Finally, a summary and possible future directions will be described in Section 7. The prolonged proofs are postponed to the Appendix A.

## 2. Problem Formulation and Basic Properties

The DSIB function is a multi-terminal extension of the standard IB [1]. First, we briefly remind the latter’s definition and give related results that will be utilized for its double-sided counterpart. Then, we provide a proper definition of the DSIB optimization problem and present some general preliminaries.

### 2.1. The Single-Sided Information Bottleneck (SSIB) Function

**Definition** **1**(SSIB). *Let (X,V) be a pair of random variables with |X|=n, |V|=m, and fixed $P_{XV}$. Denote by q the marginal probability vector of X, and let T be the transition matrix from X to V, i.e.,*
$$T_{ij}≜P\left(V=i|X=j\right),\;1\leq i\leq m,\;1\leq j\leq n.$$
*Consider all random variables U satisfying the Markov chain U→X→V. The SSIB function is defined as:*

(6)
$$\begin{array}{cccccc} \; & {\hat{R}}_{T}\left(q,C\right)≜\; & \; & \underset{P_{U|X}}{\mathrm{maximize}}\; & \; & I(U;V)\\ \; & \; & \; & \mathrm{subject}\;\mathrm{to}\; & \; & I(X;U)\leq C. \end{array}$$



**Remark** **1.**
*The SSIB problem defined in (Equation 6) is equivalent (has similar solution) to the CEB problem considered in [17].*


Although the optimization problem in (Equation 6) is well defined, the auxiliary random variable U may have an unbounded alphabet. The following lemma provides an upper bound on the cardinality of U, thus relaxing the optimization domain.

**Lemma** **1**(Lemma 2.2 of [17]). *The optimization over U in (Equation 6) can be restricted to |U|≤n+1.*

**Remark** **2.**
*A tighter bound, namely |U|≤n, was previously proved in [63] for the corresponding dual problem, namely, the IB Lagrangian. However, since ${\hat{R}}_{T}\left(q,C\right)$ is generally not a strictly convex function of C, it cannot be directly applied for the primal problem (Equation 6).*


Note that the SSIB optimization problem (Equation 6) is basically a convex function maximization over a convex set; thus, the maximum is attained on the boundary of the set.

**Lemma** **2**(Theorem 2.5 of [17]). *The inequality constraint in (Equation 6) can be replaced by equality constraint, i.e., I(X;U)=C.*

### 2.2. The Double-Sided Information Bottleneck (DSIB) Function

**Definition** **2**(DSIB). *Let (X,Y) be a pair of random variables with |X|=n, |Y|=m and fixed $P_{XY}$. Consider all the random variables U and V satisfying the Markov chain U→X→Y→V. The DSIB function $R:\left[0,H\left(X\right)\right]\times \left[0,H\left(Y\right)\right]\to R_{+}$ is defined as:*
(7)$$\begin{array}{cccccc} \; & R_{P_{XY}4pt}\left(C_{u},C_{v}\right)≜\; & \; & \underset{P_{U|X},P_{V|Y}}{\mathrm{maximize}}\; & \; & I(U;V)\\ \; & \; & \; & \mathrm{subject}\;\mathrm{to}\; & \; & I\left(X;U\right)\;\leq \;C_{u}\;\mathrm{and}\;I\left(Y;V\right)\;\leq \;C_{v}. \end{array}$$
*The achieving conditional distributions $P_{U|X}$ and $P_{V|Y}$ will be termed as the optimal test-channels. Occasionally, we will drop the subscript denoting the particular choice of the bivariate source $P_{XY}$.*


Note that (Equation 7) can be expressed in the following equivalent form:(8)$$\begin{array}{ccccccc} \; & R(C_{u},C_{v})≜\;\; & \; & \underset{P_{V|Y}}{\mathrm{maximize}}\; & \; & \underset{P_{U|X}}{\mathrm{maximize}}\; & I(U;V).\\ \; & \; & \; & \mathrm{subject}\;\mathrm{to}\; & \; & \mathrm{subject}\;\mathrm{to}\\ \; & \; & & I\left(Y;V\right)\leq C_{v}\; & \; & I\left(X;U\right)\leq C_{u} \end{array}$$

Evidently, we can define (Equation 8) using (Equation 6). Indeed, fix $P_{V|Y}$ so that it satisfies $I\left(Y;V\right)\leq C_{v}$. Denote by $T_{V|Y}$ the transition matrix from Y to V and by $T_{Y|X}$ the transition matrix from X to Y, respectively, i.e.,
$$\begin{array}{cc} {\left(T_{V|Y}\right)}_{ik} & ≜P\left(V=i|Y=k\right),\;1\leq i\leq \left|V\right|,1\leq k\leq m,\\ {\left(T_{Y|X}\right)}_{kj} & ≜P\left(Y=k|X=j\right),\;1\leq k\leq m,1\leq j\leq n. \end{array}$$

Denote by $q_{x}$ and $q_{y}$ the marginal probability vectors of X and Y, respectively, and consider the inner maximization term in (Equation 8). Since $P_{V|Y}$ and $P_{XY}$ are fixed, then $P_{XV}=\sum_{y}P_{V|Y}\left(\;\cdot \;|y\right)P_{XY}\left(\;\cdot \;,y\right)$ is also fixed. Denote by $T_{V|X}≜T_{V|Y}T_{Y|X}$ the transition matrix from X to V. Therefore, the inner maximization term in (Equation 8) is just the SSIB function with parameters $T_{V|X}$ and $C_{u}$, namely, ${\hat{R}}_{T_{V|X}}\left(q_{x},C_{u}\right)$. Hence, our problem can also be interpreted in the following two equivalent ways:(9)$$\begin{array}{cccccc} \; & R(C_{u},C_{v})≜\; & \; & \underset{T_{V|Y}}{\mathrm{maximize}}\; & \; & {\hat{R}}_{T_{V|Y}T_{Y|X}}\left(q_{x},C_{u}\right)\\ \; & \; & \; & \mathrm{subject}\;\mathrm{to}\; & \; & I\left(q_{y},T_{V|Y}\right)\leq C_{v}; \end{array}$$
or, similarly, by interchanging the order of maximization in (Equation 8), it can be expressed as follows:(10)$$\begin{array}{cccccc} \; & R(C_{u},C_{v})≜\; & \; & \underset{T_{U|X}}{\mathrm{maximize}}\; & \; & {\hat{R}}_{T_{U|X}T_{X|Y}}\left(q_{y},C_{v}\right)\\ \; & \; & \; & \mathrm{subject}\;\mathrm{to}\; & \; & I\left(q_{x},T_{U|X}\right)\leq C_{u}, \end{array}$$
where $T_{U|X}$ is the transition matrix from X to U, and $T_{X|Y}$ is the transition matrix from Y to X. This representation gives us a different perspective on our problem as an optimal compressed representation of the relevance random variable for the IB framework.

**Remark** **3.**
*Taking $C_{v}=\infty$ in (Equation 9) results in an deterministic channel from Y to V, i.e., V=Y. Thus, the DSIB problem defined in (Equation 7) reduces to the SSIB problem (Equation 6).*


The bound from Lemma 1 can be utilized to give cardinality bounding for the double-sided problem.

**Proposition** **1.**
*For the DSIB optimization problem defined in (Equation 7), it suffices to consider random variables U and V with cardinalities |U|≤n+1 and |V|≤m+1.*


**Proof.** Let $T_{U|X}$ and $T_{V|Y}$ be two arbitrary transition matrices. By Lemma 1, there exists $T_{\tilde{U}|X}$ with $|\tilde{U}|\leq n+1$ such that $I\left(\tilde{U};V\right)\geq I\left(U;V\right)$ and $I\left(X;\tilde{U}\right)\leq C_{u}$. Similarly, $T_{V|Y}$ can be replaced with $T_{\tilde{V}|Y}$, $|\tilde{V}|\leq m+1$ such that $I\left(\tilde{U};\tilde{V}\right)\geq I\left(\tilde{U},V\right)\geq I\left(U;V\right)$, and $I\left(Y;\tilde{V}\right)\leq C_{v}$. Therefore, there exists an optimal solution with |U|≤n+1 and |V|≤m+1.    □

In the following two sections, we will present the primary analytical outcomes of our study. First, we consider the scenario where our bivariate source is binary, specifically DSBS. Then, we handle the case where X and Y are jointly Gaussian.

## 3. Binary (X,Y)

Let (X,Y) be a DSBS with parameter *p*, i.e.,
(11)$$P_{XY}\left(x,y\right)=\frac{1}{2}\left(p\;\cdot \;𝟙\left(x\neq y\right)+\left(1-p\right)𝟙\left(x=y\right)\right).$$

We entitle the respective optimization problem (Equation 7) as the *binary double-sided information bottleneck* (BDSIB) and emphasize its dependence on the parameter *p* as $R(C_{u},C_{v},p)$.

The following proposition states that the cardinality bound from Lemma 1 can be tightened in the binary case.

**Proposition** **2.**
*Considering the optimization problem in (Equation 6) with X=Ber(q) and |Y|=3, binary U is optimal.*


The proof of this proposition is postponed to Appendix A. Using similar justification for Proposition 1 combined with Proposition 2, we have the following strict cardinality formula for the BDSIB setting.

**Proposition** **3.**
*For the respective DSBS setting of (Equation 7), it suffices to consider random variables U and V with cardinalities |U|=|V|=2.*


Note that the above statement is not required for the results in the rest of this section to hold and will be mainly applied to justify our conjectures via numerical simulations.

We next show that the specific objective function for the binary setting of (Equation 7), i.e, the mutual information between U and V, has an elegant representation which will be useful in deriving lower and upper bounds.

**Lemma** **3.**
*The mutual information between U and V can be expressed as follows:*

(12)
$$I\left(U;V\right)=E_{P_{U}\times P_{V}}\left[K(U,V,p)\log K(U,V,p)\right],$$

*where the expectation is taken over the product measure $P_{U}\times P_{V}$, U and V are binary random variables satisfying:*

(13)
$$P\left(U=0\right)=\frac{\alpha_{1}-\frac{1}{2}}{\alpha_{1}-\alpha_{0}},\;P\left(V=0\right)=\frac{\beta_{1}-\frac{1}{2}}{\beta_{1}-\beta_{0}},$$

*the kernel K(u,v,p) is given by:*

(14)
$$K\left(u,v,p\right)=2\alpha_{u}*\beta_{v}*p=1-\left(1-2p\right)\left(1-2\alpha_{u}\right)\left(1-2\beta_{v}\right),$$

*and the reverse test-channels are defined by: $\alpha_{u}≜P\left(X=1|U=u\right)$, $\beta_{v}≜P\left(Y=0|V=v\right)$. Furthermore, since $|\left(1-2p\right)\left(1-2\alpha_{u}\right)\left(1-2\beta_{v}\right)|<1$, utilizing Taylor’s expansion of log(1−x), we obtain:*

(15)
$$I\left(U;V\right)=\sum_{n=2}^{\infty }\frac{{\left(1-2p\right)}^{n}E_{}\left[{\left(1-2\alpha_{U}\right)}^{n}\right]E_{}\left[{\left(1-2\beta_{V}\right)}^{n}\right]}{n(n-1)}.$$



The general cascade of test-channels and the DSBS, defined by ${\left\{\alpha_{u}\right\}}_{u=0}^{1}$, ${\left\{\beta_{v}\right\}}_{v=0}^{1}$ and *p*, is illustrated in Figure 9. The proof of Lemma 3 is postponed to Appendix B.

We next examine some corner cases for which $R(C_{u},C_{v},p)$ is fully characterized.

### 3.1. Special Cases

A particular case where we have a complete analytical solution is when *p* tends to 1/2.

**Theorem** **1.**
*Suppose $p=\frac{1}{2}-\epsilon$, and consider ϵ→0. Then*

(16)
$$R\left(C_{u},C_{v},\epsilon \right)=2\epsilon^{2}\log e\;\cdot \;{\left(1-2h_{b}^{-1}\left(1-C_{u}\right)\right)}^{2}{\left(1-2h_{b}^{-1}\left(1-C_{v}\right)\right)}^{2}+o\left(\epsilon^{2}\right),$$

*and it is achieved by taking $P_{U|X}$ and $P_{V|Y}$ as BSC test-channels satisfying the constraints with equality.*


This theorem follows by considering the low SNR regime in Lemma 3 and is proved in Appendix D. For the lower bound we take $P_{U|X}$ and $P_{V|Y}$ to be BSCs.

In Section 6 we will give a numerical evidence that BSC test-channels are in fact optimal provided that *p* is sufficiently large. However, for small *p* this is no longer the case and we believe the following holds.

**Conjecture** **1.**
*Let X=Y, i.e., p=0. The optimal test-channels $P_{U|X}$ and $P_{V|X}$ that achieve $R(C_{u},C_{v},0)$ are Z-channel and S-channel respectively.*


**Remark** **4.**
*Our results in the numerical section strongly support this conjecture. In fact they prove it within the resolution of the experiments, i.e., for optimizing over a dense set of test-channels rather then all test-channels. Nevertheless, we were not able to find an analytical proof for this result.*


**Remark** **5.**
*Suppose X=Y, $I\left(X;U\right)=C_{u}$, and $I\left(X;V\right)=C_{v}$. Since I(U;V)=I(U;X)+I(V;X)−I(X;U,V) (as U→X→Y→V form a Markov chain in this order) then maximizing I(U;V) is equivalent to minimizing I(X;U,V), namely, minimizing information combining as in [23,35]. Therefore, Conjecture 1 is equivalent to the conjecture that among all channels with $I\left(X;U\right)\geq C_{u}$ and $I\left(Y;V\right)\geq C_{v}$, Z and S are the worst channels for information combining.*


This observation leads us the following additional conjecture.

**Conjecture** **2.**
*The test-channels $P_{U|X}$ and $P_{V|X}$ that maximize I(X;U,V) are both Z channels.*


**Remark** **6.**
*Suppose now that p is arbitrary and assume that one of the channels $P_{U|X}$ or $P_{V|Y}$ is restricted to be a binary memoryless symmetric (BMS) channel (Chapter 4 of [64]), then the maximal I(U;V) is attained by BSC channels, as those are the symmetric channels minimizing I(X;U,V) [23]. It is not surprising that once the BMS constraint is removed, symmetric channels are no longer optimal (see the discussion in (Section VI.C of [23])).*


Consider now the case X=Y (p=0) with an additional symmetry assumption $C_{u}=C_{v}$. The most reasonable apriori guess is that the optimal test-channels $P_{U|X}$ and $P_{V|X}$ are the same up to some permutation of inputs and outputs. Surprisingly, this is not the case, unless they are BSC or Z channels, as the following negative result states.

**Proposition** **4.**
*Suppose $C_{u}=C_{v}$ and the transition matrix from X to V, given by*

(17)
$$T_{V|X}=\left(\begin{array}{cc} a & b\\ 1-a & 1-b \end{array}\right),$$

*satisfies $I\left(u_{2},T_{V|X}\right)=C_{v}$. Consider the respective SSIB optimization problem*

(18)
$${\hat{R}}_{T_{V|X}}\left(u_{2},C_{u}\right)=\max_{P_{U|X}:I\left(U;X\right)\leq C_{u}}I\left(U;V\right).$$


*The optimal $P_{U|X}$ that attains (Equation 6) with $q_{X}=u_{2}$ and $C=C_{u}$ does not equal to $P_{V|X}$ or any permutation of $P_{V|X}$, unless $P_{V|X}$ is a BSC or a Z channel.*


The proof is based on [17] and is postponed to Appendix E.

As for the case of X≠Y, i.e., p≠0, we have the following conjecture.

**Conjecture** **3.**
*For every $\left(C_{u},C_{v}\right)\in \left[0,1\right]\times \left[0,1\right]$, there exists $\theta (C_{u},C_{v})$, such that for every $p>\theta (C_{u},C_{v})$ the achieving test-channels $P_{U|X}$ and $P_{V|Y}$ are BSC with parameters $\alpha =h_{b}^{-1}\left(1-C_{u}\right)$ and $\beta =h_{b}^{-1}\left(1-C_{v}\right)$ respectively.*


We will provide supporting arguments for this conjecture via numerical simulations in Section 6.

### 3.2. Bounds

In this section we present our lower and upper bounds on the BDSIB function, then we compare them for various channel parameters. The proofs are postponed to Appendix F. For the simplicity of the following presentation we define
(19)$$g_{b}\left(x\right)≜\frac{1}{2(1-x)}h_{b}\left(x\right),\;x\in \left[0,1/2\right],$$
denote $g_{b}^{-1}\left(\;\cdot \;\right)$ as its inverse restricted to [0,1], and ℏ(x)≜−xlogx.

**Proposition** **5.**
*The BDSIB function is bounded from below by*

(20)
$$\begin{array}{c} \;R(C_{u},C_{v},p)\geq \\ \max\left\{\begin{array}{cc} & 1-h_{b}\left(\alpha *\beta *p\right),\\ & 1-\frac{1}{2\bar{\delta }\bar{\zeta }}\left[\hbar \left(\delta *\zeta *p\right)+\left(1-2\zeta \right)\;\cdot \;\hbar \left(\bar{\delta }*p\right)+\left(1-2\delta \right)\;\cdot \;\hbar \left(\bar{\zeta }*p\right)+\left(1-2\delta \right)\left(1-2\zeta \right)\;\cdot \;\hbar \left(p\right)\right], \end{array}\right. \end{array}$$

*where $\alpha =h_{b}^{-1}\left(1-C_{u}\right)$, $\beta =h_{b}^{-1}\left(1-C_{v}\right)$, $\delta =g_{b}^{-1}\left(1-C_{u}\right)$, and $\zeta =g_{b}^{-1}\left(1-C_{v}\right)$.*


All terms in the RSH of (Equation 20) are attained by taking test-channels that match the constraints with equality and plugging them in Lemma 3. In particular: the first term is achieved by BSC test channels with transition probabilities α and β; the second term is achieved by taking $P_{U|X}$ be a Z(δ) channel and $P_{V|Y}$ be an S(ζ) channel. The aforementioned test-channel configurations are illustrated in Figure 10.

We compare the different lower bounds derived in Proposition 5 for various values of constraints. The achievable rate vs channel transition probability *p* is shown in Figure 11. Our first observation is that BSC test-channels outperform all other choices for almost all values of *p*. However, Figure 12 gives a closer look on small values of *p*. It is evident that the combination of Z and S test-channels outperforms any other schemes for small values of *p*. We have used this observation as one supporting evidence to Conjecture 1.

We proceed to give an upper bound.

**Proposition** **6.**
*A general upper bound on BDSIB is given by*

(21)
$$R\left(C_{u},C_{v},p\right)\leq \min\left\{\begin{array}{cc} \; & {\left(1-2p\right)}^{2}(1-2h_{b}^{-1}{\left(1-C_{u}\right)}^{2}(1-2h_{b}^{-1}{\left(1-C_{v}\right)}^{2},\\ \; & \min\{1-h_{b}\left(h_{b}^{-1}\left(1-C_{u}\right)*p\right),1-h_{b}\left(h_{b}^{-1}\left(1-C_{v}\right)*p\right)\}. \end{array}\right.$$



Note that the first term can be derived by applying Jensen’s inequality on (Equation 12), and the second term is a combination of the standard IB and the cut-set bound. We postpone the proof of Proposition 6 to Appendix F.

**Remark** **7.**
*Since $p=\frac{1}{2}-\epsilon$, we have a factor 2 loss in the first term compared to the precise behavior we have found for $p\approx \frac{1}{2}$ in Theorem 1. This loss comes from the fact that the bound in (Equation 21) actually upper bounds the χ-squared mutual information between U and V. It is well-known that for very small I(X;Y) we have that $I\left(X;Y\right)\approx 1/2I_{\chi^{2}}\left(X;Y\right)$, see [65].*


We compare the different upper bounds from Proposition 6 in Figure 13 for various bottleneck constraints, and in Figure 14 for various values of channel transition probabilities *p*. We observe that there are regions of *C* and *p* for which Jensen’s based bound outperforms the standard IB bound.

Finally, we compare the best lower and upper bounds from Propositions 5 and 6 for various values of channel parameters in Figure 15. We observe that the bounds are tighter for asymmetric constraints and high transition probabilities.

## 4. Gaussian (X,Y)

In this section we consider a specific setting where (X,Y) is the normalized zero mean Gaussian bivariate source, namely,
(22)$$\left(\begin{array}{c} X\\ Y \end{array}\right)∼N\left(\left(\begin{array}{c} 0\\ 0 \end{array}\right),\left(\begin{array}{cc} 1 & \rho \\ \rho & 1 \end{array}\right)\right).$$
We establish achievability schemes and show that Gaussian test-channels $P_{U|X}$ and $P_{V|Y}$ are optimal for vanishing SNR. Furthermore we show an elegant representation of the problem through *probabilistic Hermite polynomials* which are defined by
(23)$$H_{n}\left(x\right)≜{\left(-1\right)}^{n}e^{\frac{x^{2}}{2}}\frac{d^{n}}{dx^{n}}e^{-\frac{x^{2}}{2}},\;n\in N_{0}.$$

We denote the Gaussian DSIB function with explicit dependency on ρ as $R(C_{u},C_{v},\rho )$.

**Proposition** **7.**
*Let $H_{n}\left(\;\cdot \;\right)$ be the nth order probabilistic Hermite polynomial, then the objective function of (Equation 7) for the Gaussian setting is given by*

(24)
$$I\left(\;U\;;\;V\;\right)\;=\;E_{UV}\;\left[\;\log\;\left(\;\sum_{n=0}^{\infty }\;\frac{\rho^{n}}{n!}\;E_{}\left[H_{n}\left(X\right)|U\right]E_{}\left[H_{n}\left(Y\right)|V\right]\;\right)\;\right].$$



This representation follows by considering $I\left(U;V\right)=D(P_{UV}||P_{U}\;\cdot \;P_{V})$ and expressing $\frac{P_{UV}}{P_{U}\;\cdot \;P_{V}}$ using Mehler Kernel [66]. Mehler Kernel decomposition is a special case of a much richer family of Lancaster distributions [67]. The proof of Proposition 7 is relegated to Appendix G.

Now we give two lower bounds on $R(C_{u},C_{v},\rho )$. Our first lower bound is established by choosing $P_{U|X}$ and $P_{V|Y}$ to be additive Gaussian channels, satisfying the mutual information constraints with equality.

**Proposition** **8.**
*A lower bound on $R(\;C_{u}\;,\;C_{v}\;,\rho \;)$ is given by*

(25)
$$R\left(C_{u},C_{v},\rho \right)\;\geq \;-\frac{1}{2}\log\;\left(\;1\;-\;\rho^{2}\left(\;1\;-\;2^{-2C_{u}}\;\right)\left(\;1\;-\;2^{-2C_{v}}\;\right)\;\right).$$



The proof of this bound is developed in Appendix H.

Although it was shown in [26] that choosing the test-channel to be Gaussian is optimal for the single-sided variant, it is not the case for its double-sided extension. We will show this by examining a specific set of values for the rate constraints, $\left(C_{u},C_{v}\right)=\left(1,1\right)$. Furthermore, we choose the test channels $P_{U|X}$ and $P_{V|Y}$ to be deterministic quantizers.

**Proposition** **9.**
*Let $\left(C_{u},C_{v}\right)=\left(1,1\right)$, then*

(26)
$$R\left(1,1,\rho \right)\geq 1-h_{2}\left(\frac{\arccos\rho }{\pi }\right).$$



The proof of this bound is developed in Appendix I.

We compare the bounds from Propositions 8 and 9 with $\left(C_{u},C_{v}\right)=\left(1,1\right)$ in Figure 16. The most unexpected observation here is that the deterministic quantizers lower bound outperform the Gaussian test-channels for high values of ρ. The crossing point of those bounds is given by
(27)$$\rho_{\mathrm{cros}}=\frac{e}{\sqrt{1+e^{2}}}\to \sqrt{SNR_{\mathrm{cros}}}=\frac{\rho_{\mathrm{cros}}}{\sqrt{1-\rho_{\mathrm{cros}}^{2}}}=e.$$

We proceed to present our upper bound on $R(C_{u},C_{v},\rho )$. This bound is a combination of the cutset bound and the single-sided Gaussian IB.

**Proposition** **10.**
*An upper bound on (Equation 7) with Gaussian (X,Y) setting (Equation 22) is given by*

(28)
$$R\left(C_{u},C_{v},\rho \right)\leq \min\left\{-\frac{1}{2}\log\left(1-\rho^{2}\left(1-2^{-2C_{u}}\right)\right),-\frac{1}{2}\log\left(1-\rho^{2}\left(1-2^{-2C_{v}}\right)\right)\right\}.$$



We compare the best lower and upper bounds from Propositions 8–10 in Figure 17. We observe that the bounds become tighter as the constraint increases and in the low-SNR regime.

### 4.1. Low-SNR Regime

For ρ→0, the exact asymptotic behavior of the Gaussian (Proposition 8) and deterministic (Proposition 9) test-channels, respectively, for $C_{u}=C_{v}=1$ is given by:$$\begin{array}{cc} \lim_{\rho \to 0}\;-\;\frac{1}{2}\;\log\;\left(\;1\;-\;\rho^{2}\;\left(1\;-\;2^{-2C_{u}}\;\right)\left(1\;-\;2^{-2C_{v}}\;\right)\right)\; & \;=\;\frac{9\log e}{32}\rho^{2}\;+\;o\left(\;\rho^{2}\;\right),\\ \lim_{\rho \to 0}1-h_{2}\left(\frac{\arccos\rho }{\pi }\right) & \;=\;\frac{2\log e}{\pi^{2}}\rho^{2}\;+\;o\left(\;\rho^{2}\;\right). \end{array}$$
Hence, the Gaussian choice outperforms the second lower bound for vanishing SNR. The following theorem establishes that Gaussian test-channels are optimal for low-SNR.

**Theorem** **2.**
*For small ρ, the GDSIB function is given by:*

(29)
$$R\;\left(\;C_{u},\;C_{v},\;\rho \;\right)\;=\;\frac{\rho^{2}\;\log\;e}{2}\left(\;1\;-\;2^{-2C_{u}}\;\right)\;\left(\;1\;-\;2^{-2C_{v}}\;\right)\;+\;o\left(\;\rho^{2}\;\right).$$



The lower bound follows from Proposition 8. The upper bound is established by considering the kernel representation from Proposition 7 in the limit of vanishing ρ. The detailed proof is given in Appendix J.

### 4.2. Optimality of Symbol-by-Symbol Quantization When X=Y

Consider an extreme scenario for which X=Y∼N(0,1). Taking the encoders $P_{U|X}$ and $P_{V|X}$ as a symbol-by-symbol deterministic quantizers satisfying:$$H\left(U\right)=H\left(V\right)=\min\left\{C_{u},C_{v}\right\},$$
we achieve the optimum
$$I\left(U;V\right)=\min\left\{C_{u},C_{v}\right\}.$$

## 5. Alternating Maximization Algorithm

Consider the DSIB problem for DSBS with parameter *p* analyzed in Section 3. The respective optimization problem involves simultaneous search of the maximum over the sets $\left\{P_{U|X}\right\}$ and $\left\{P_{V|Y}\right\}$. An alternating maximization, namely, fixing $P_{U|X}$, then finding the respective optimal $P_{V|Y}$ and vice versa, is sub-optimal in general and may result in convergent to a saddle point. However, for the case p=0 with symmetric bottleneck constraints, Proposition 4 implies that such point exists only for the BSC and Z/S channels. This motivates us to believe that performing an alternating maximization procedure on (Equation 9) will not result in sub-optimal saddle point, but rather converge to the optimal solution also for the general discrete (X,Y).

Thus, we propose an alternating maximization algorithm. The main idea is to fix $P_{V|Y}$ and then compute $P_{U|X}^{*}$ that attains the inner term in (Equation 9). Then, using $P_{U|X}^{*}$, we find the optimal $P_{V|Y}^{*}$ that attains the inner term in (Equation 10). Then, we repeat the procedure in alternating manner until convergence.

Note that inner terms of (Equation 9) and (Equation 10) are just the standard IB problem defined in (Equation 6). For completeness, we state here the main result from [1] and adjust it for our problem. Consider the respective Lagrangian of (Equation 6) given by:(30)$$L\left(P_{U|X},\lambda \right)=I\left(U;V\right)+\lambda \left(C-I\left(X;U\right)\right).$$

**Lemma** **4**(Theorem 4 of [1]). *The optimal test-channel that maximizes (Equation 30) satisfies the equation:*
(31)$$P_{U|X}\left(u|x\right)=\frac{P_{U}\left(u\right)}{Z(x,\beta )}e^{-\beta D(P_{V|X=x}∥P_{V|U=u})},$$
*where β≜1/λ and $P_{V|U}$ is given via Bayes’ rule, as follows:*
(32)$$P_{V|U}\left(v|u\right)=\frac{1}{P_{U}\left(u\right)}\sum_{x}P_{V|X}\left(v|x\right)P_{U|X}\left(u|x\right)P_{X}\left(x\right).$$

In a very similar manner to the Blahut–Arimoto algorithm [18], the self-consistent equations can be adapted into converging, alternating iterations over the convex sets $\left\{P_{U|X}\right\}=\Delta_{n}^{⊗n}$, $\left\{P_{U}\right\}=\Delta_{n}$, and $\left\{P_{V|U}\right\}=\Delta_{n}^{⊗n}$, as stated in the following lemma.

**Lemma** **5**(Theorem 5 of [1]). *The self-consistent equations are satisfied simultaneously at the minima of the functional:*
(33)$$F\left(P_{U|X},P_{U},P_{Y|U}\right)=I\left(U;X\right)+\beta E_{}\left[D(P_{V|X}∥P_{V|U})\right],$$
*where the minimization is performed independently over the convex sets of $\left\{P_{U|X}\right\}=\Delta_{n}^{⊗n}$, $\left\{P_{U}\right\}=\Delta_{n}$, and $\left\{P_{V|U}\right\}=\Delta_{n}^{⊗n}$. The minimization is performed by the converging alternation iterations as described in Algorithm 1.*

**Algorithm 1:** IB iterative algorithm IBAM(args)

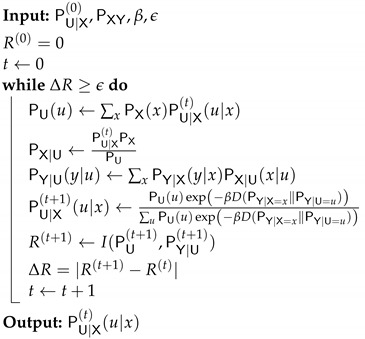



Next, we propose a combined algorithm to solve the optimization problem from (Equation 7). The main idea is to fix one of the test-channels, i.e., $P_{V|Y}$, and then find the corresponding optimal opposite test-channel, i.e., $P_{U|X}$, using Algorithm 1. Then, we apply again Algorithm 1 by switching roles, i.e., fixing the opposite test-channel, i.e., $P_{U|X}$, and then identifying the optimal $P_{V|Y}$. We repeat this procedure until convergence of the objective function I(U;V). We summarize the proposed composite method in Algorithm 2.

**Remark** **8.**
*Note that every alternating step of the algorithm involves finding an optimal $\left(\beta^{*},\eta^{*}\right)$ that corresponds to the respective problem constraints $\left(C_{u},C_{v}\right)$. We have chosen to implement this exploration step using a bisection-type method. This may result that the actual pair $\left(C_{u},C_{v}\right)$ is ϵ-far away from the desired constraint.*


**Algorithm 2:** DSIB iterative algorithm DSIBAM(args)

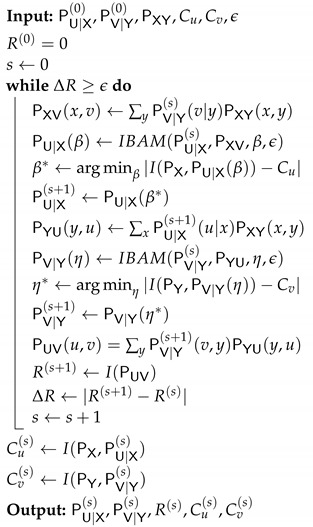



## 6. Numerical Results

In this section, we focus on the DSBS setting of Section 3. In the first part of this section, we will show using a brute-force method the existence of a sharp, phase-transition phenomena in the optimal test-channels $P_{U|X}$ and $P_{V|Y}$ vs. DSBS parameter *p*. In the second part of this section, we will evaluate the alternating maximization algorithm proposed in Section 5; then, we compare its performance to the brute-force method.

### 6.1. Exhaustive Search

In this set of simulations, we again fix the transition matrix from Y to V characterized by the parameters:(34)$$T=\left(\begin{array}{cc} a & b\\ 1-a & 1-b \end{array}\right),$$
chosen such that $I\left(Y;V\right)=C_{v}$. This choice defines a path b=f(a) in the (a,b) plain. Then, for every such *T* we optimize I(U;V) for different values of the DSBS parameter *p*. The results for a specific choice of $\left(C_{u},C_{v}\right)=\left(0.4,0.6\right)$ vs. *a* for different values of *p* are plotted in Figure 18. Note that the region of *a* corresponds to the continuous conversion from a Z channel (a=0) to a BSC ($a=a_{\max}$). We observe here a very sharp transition from the optimality of Z-S channels to BSC channels configuration for a small change in *p*. This kind of behavior continues to hold with a different choice of $\left(C_{u}=0.1,C_{v}=0.9\right)$, as can be seen in Figure 19.

Next, we would like to emphasize this sharp phase transition phenomena by plotting the optimal *a* that achieves the maximal I(U;V) vs the DSBS parameter *p*. The results for various combinations of $C_{u}$ and $C_{v}$ are presented in Figure 20 and Figure 21. We observe that the curves are convex for $p\in [0,p_{th})$ and constant for $p>p_{th}$ with $a=a_{bsc}$. Furthermore, the derivative of a(p) for $p\to p_{th}$ tends to *∞*.

One may further claim that there is no sharp transition to the BSC test-channels $P_{U|X}$ and $P_{V|Y}$ as *p* grows away from zero, but rather only approaches BSC. To convince the reader that the optimal test channels are exactly BSC, we performed an alternating maximization experiment. We fixed p>0, $C_{u}$ and $C_{v}$. Then we have chosen $P_{V|Y}$ as an almost BSC channel satisfying $I\left(Y;V\right)\leq C_{v}$ and found the channel $P_{X|U}$ that maximizes I(U;V) subject to $I\left(X;U\right)\leq C_{u}$. Then, we fixed the channel $P_{X|U}$ and found the $P_{Y|V}$ that maximizes I(U;V) subject to $I\left(Y;V\right)\leq C_{v}$. We have repeated this alternating maximization procedure until it converges. The transition matrices were parameterized as follows:(35)$$T_{Y|V}=\left(\begin{array}{cc} q_{0} & q_{1}\\ 1-q_{0} & 1-q_{1} \end{array}\right),\;T_{X|U}=\left(\begin{array}{cc} p_{0} & p_{1}\\ 1-p_{0} & 1-p_{1} \end{array}\right).$$

The results for different values of *p*, $C_{u}$, and $C_{v}$ are shown in Figure 22, Figure 23 and Figure 24. We observe that $p_{0}$ and $q_{0}$ rapidly converge to their respective BSC values satisfying the mutual information constraints. Note that the last procedure is still an exhaustive search, but it is performed in alternating fashion between the sets $\left\{P_{U|X}\right\}$ and $\left\{P_{V|Y}\right\}$.

### 6.2. Alternating Maximization

In this section, we will evaluate the algorithm proposed in Section 5. We focus on the DSBS setting of Section 3 with various values of problem parameters.

First, we explore the convergence behavior of the proposed algorithm. Figure 25 shows the objective function I(U;V) on every iteration step for representative fixed-channel transition parameters *p* and the constraints $C_{u}$ and $C_{v}$. We observe a slow convergence to a final value for p=0 and $C_{u}=C_{v}=0.2$, but once the constraints and the transition probability are increased, the algorithm converges much more rapidly. The non-monotonic behavior in some regimes is justified with the help of Remark 8. In Figure 26, we see the respective test-channel probabilities $\alpha_{0}+\alpha_{1}$, $1-\alpha_{0}$, $\beta_{0}+\beta_{1}$, and $1-\beta_{1}$. First, note that if $\alpha_{0}+\alpha_{1}=1$, then $P_{X|U}$ is a BSC. Similarly, if $\beta_{0}+\beta_{1}=1$, then $P_{Y|V}$ is a BSC. Second, if $1-\alpha_{0}=1$, then $P_{X|U}$ is a Z-channel. Similarly, if $1-\beta_{1}=1$, then $P_{Y|V}$ is an S-channel. We observe that for p=0, the test-channels $P_{X|U}$ and $P_{Y|V}$ converge to Z- and S-channels, respectively. As for all other settings, the test-channels converge to BSC channels.

Finally, we compare the outcome of Algorithm 2 to the optimal solution achieved by the brute-force method, namely, evaluating (Equation 12) for every $P_{U|X}$ and $P_{V|Y}$ that satisfy the problem constraints. The results for various values of channel parameters are shown in Figure 27. We observe that the proposed algorithm achieves the optimum for any DSBS parameter *p* and some representative constraints $C_{u}$ and $C_{v}$.

## 7. Concluding Remarks

In this paper, we have considered the Double-Sided Information Bottleneck problem. Cardinality bounds on the representation’s alphabets were obtained for an arbitrary discrete bivariate source. When X and Y are binary, we have shown that taking binary auxiliary random variables is optimal. For DSBS, we have shown that BSC test-channels are optimal when p→0.5. Furthermore, numerical simulations for arbitrary *p* indicate that Z -and S-channels are optimal for p=0. As for the Gaussian bivariate source, representation of I(U;V) utilizing Hermite polynomials was given. In addition, the optimality of the Gaussian test-channels was demonstrated for vanishing SNR. Moreover, we have constructed a lower bound attained by deterministic quantizers that outperforms the jointly Gaussian choice at high SNR. Note that the solution for the *n*-letter problem $\max\frac{1}{n}I\left(U;V\right)$ for $U\to X^{n}\to Y^{n}\to V$ under constraints $I\left(U;X^{n}\right)\leq nC_{u}$ and $I\left(V;Y^{n}\right)\leq nC_{v}$ does not tensorize in general. For $X^{n}=Y^{n}∼{\mathrm{Ber}}^{⊗n}\left(0.5\right)$, we can easily achieve the cut-set bound $I\left(U;V\right)/n=\min\left\{C_{u},C_{v}\right\}$. In addition, if time-sharing is allowed, the results change drastically.

Finally, we have proposed an alternating maximization algorithm based on the standard IB [1]. For the DSBS, it was shown that the algorithm converges to the global optimal solution.

## Figures and Tables

**Figure 1 entropy-24-01321-f001:**
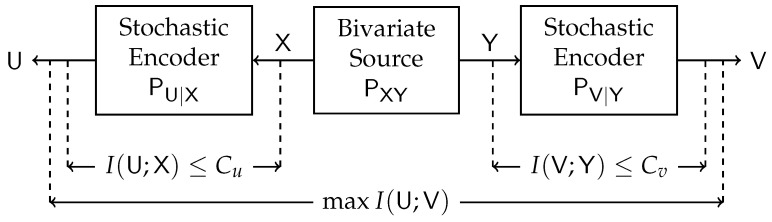
Block diagram of the Double-Sided Information Bottleneck function.

**Figure 2 entropy-24-01321-f002:**
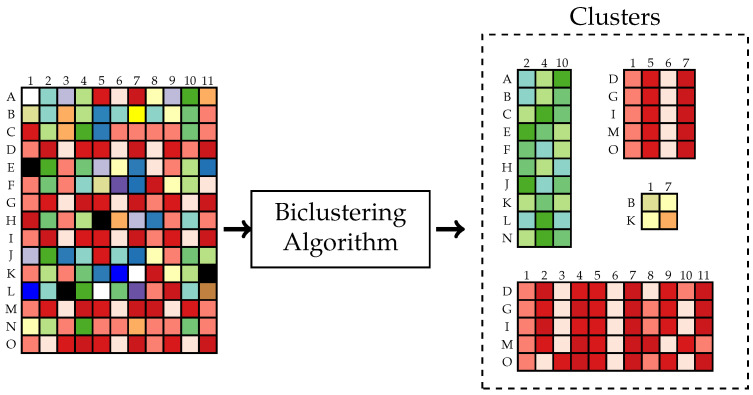
Illustration of a typical biclustering algorithm.

**Figure 3 entropy-24-01321-f003:**
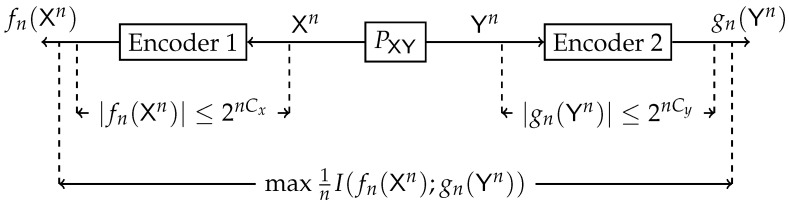
Block diagram of the information-theoretic biclustering problem.

**Figure 4 entropy-24-01321-f004:**
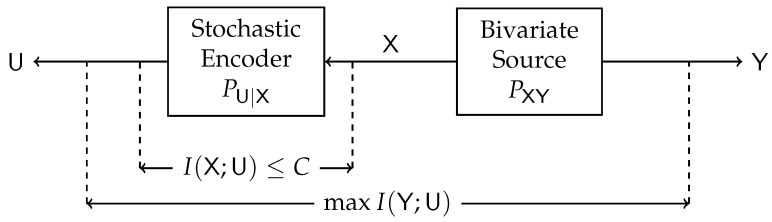
Block diagram of the Single-Sided Information Bottleneck function.

**Figure 5 entropy-24-01321-f005:**
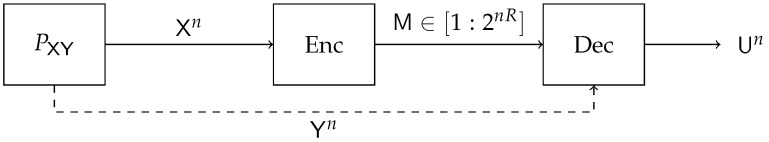
Block diagram of Source Coding with Side Information.

**Figure 6 entropy-24-01321-f006:**
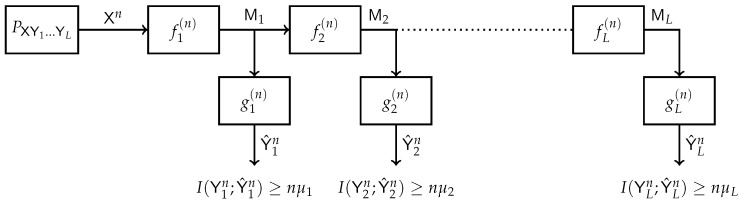
Block diagram of the Multi-Layer IB.

**Figure 7 entropy-24-01321-f007:**
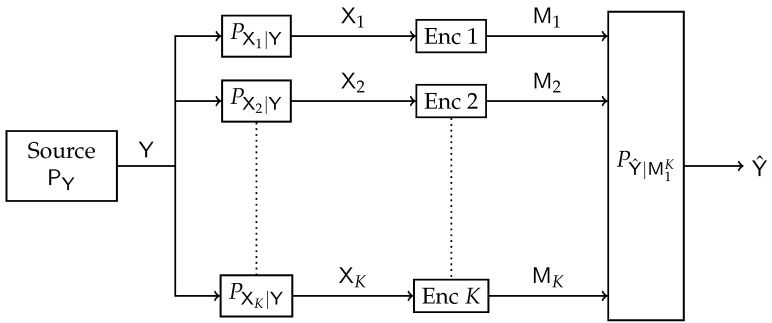
Block diagram of the Distributive IB.

**Figure 8 entropy-24-01321-f008:**
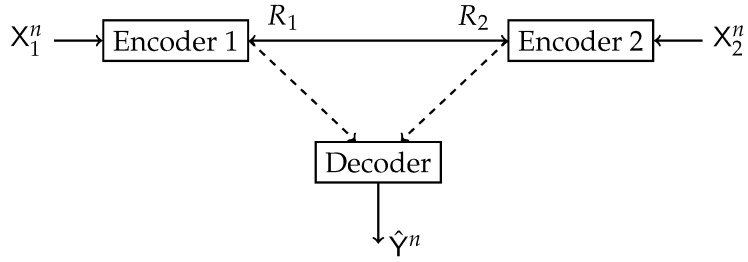
Block diagram of the Collaborative IB.

**Figure 9 entropy-24-01321-f009:**
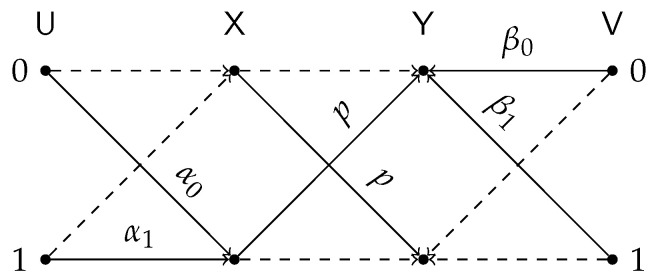
General test-channel construction of the BDSIB function.

**Figure 10 entropy-24-01321-f010:**
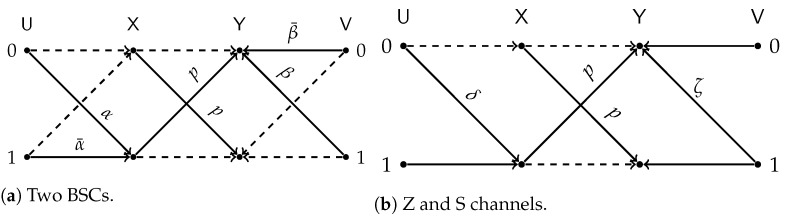
Test-channel that achieve the lower bound of Proposition 5.

**Figure 11 entropy-24-01321-f011:**
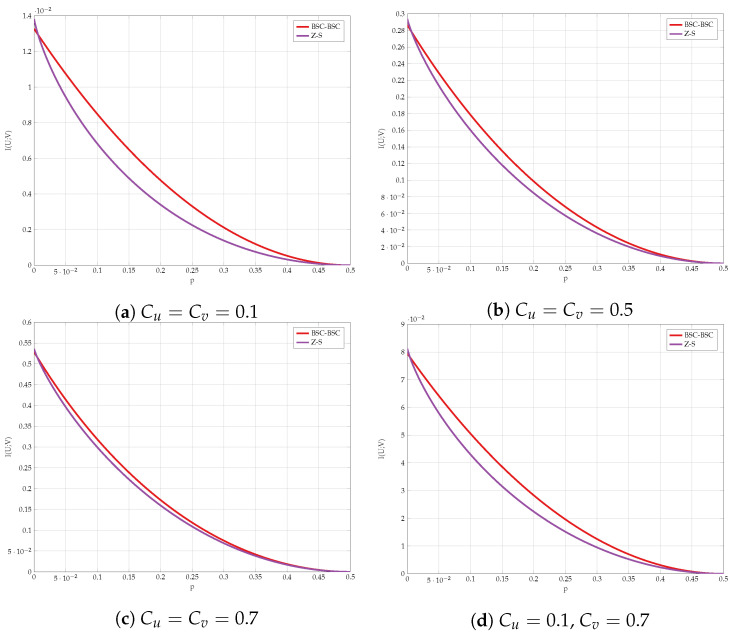
Comparison of the lower bounds.

**Figure 12 entropy-24-01321-f012:**
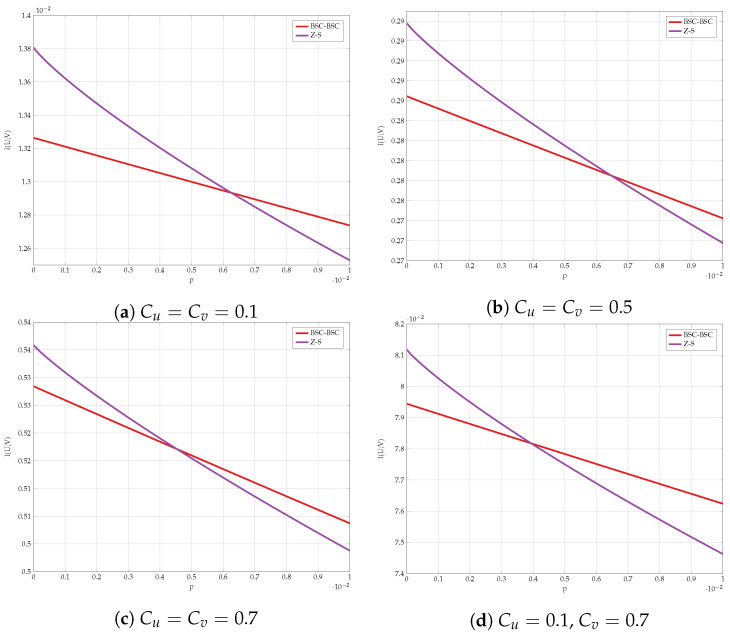
Comparison of the lower bounds in high SNR regime.

**Figure 13 entropy-24-01321-f013:**
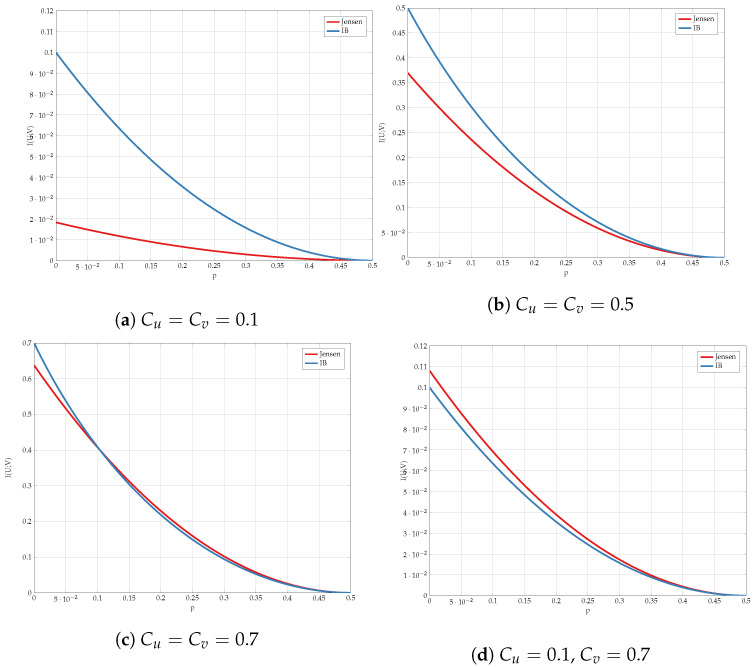
Comparison of the upper bounds for various values of $\left(C_{u},C_{v}\right)$.

**Figure 14 entropy-24-01321-f014:**
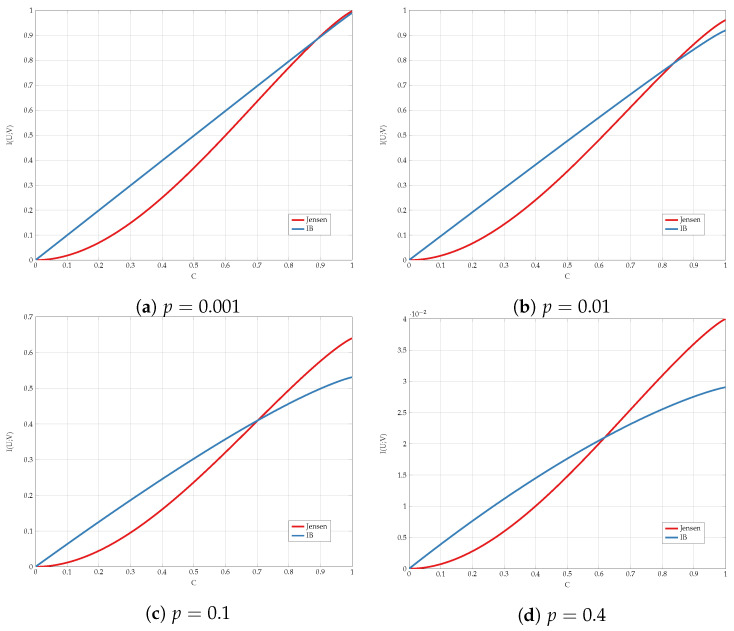
Comparison of the upper bounds for various values of *p*.

**Figure 15 entropy-24-01321-f015:**
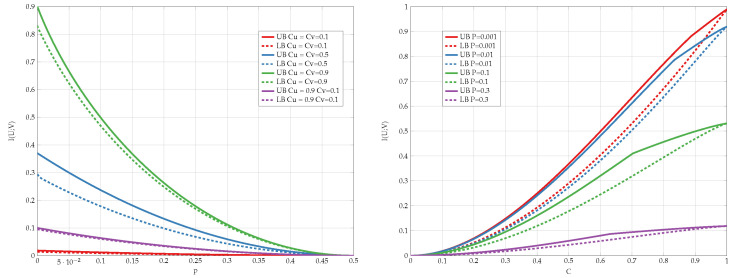
Capacity bounds for various values of *p* and $C=C_{u}=C_{v}$.

**Figure 16 entropy-24-01321-f016:**
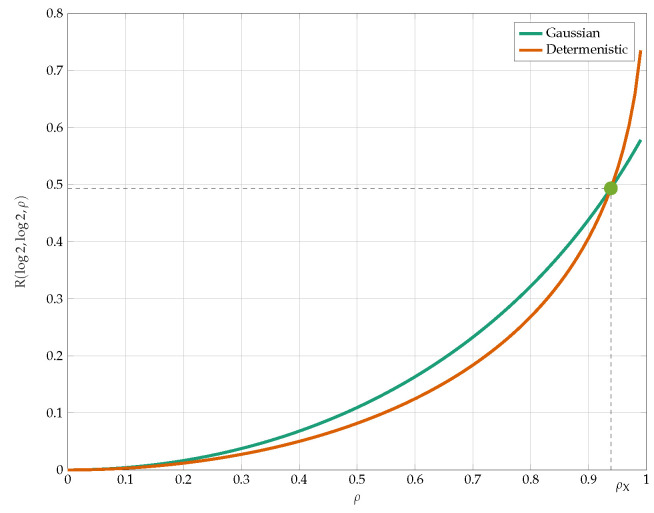
Comparison of the lower bounds from Propositions 8 and 9.

**Figure 17 entropy-24-01321-f017:**
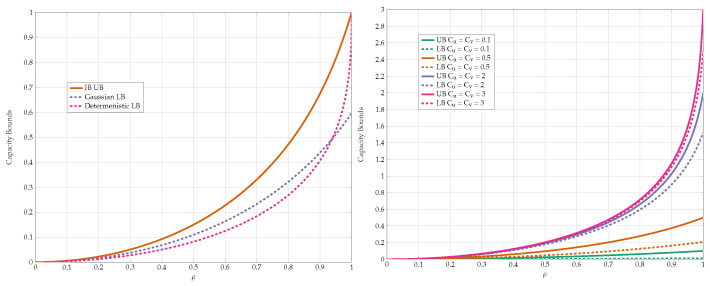
Capacity bounds for various values of *p* and $C=C_{u}=C_{v}=1$.

**Figure 18 entropy-24-01321-f018:**
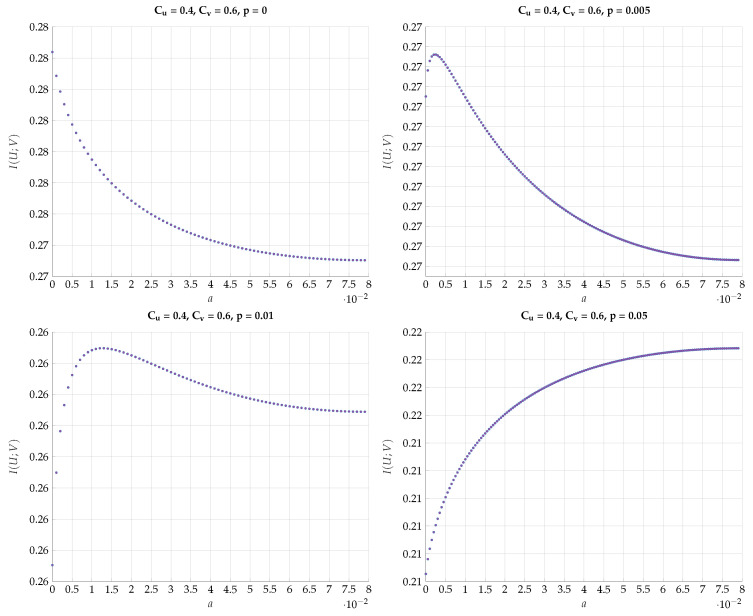
Maximal I(U;V) for fixed values $\left(C_{u},C_{v}\right)=\left(0.4,0.6\right)$ and different values of *p*.

**Figure 19 entropy-24-01321-f019:**
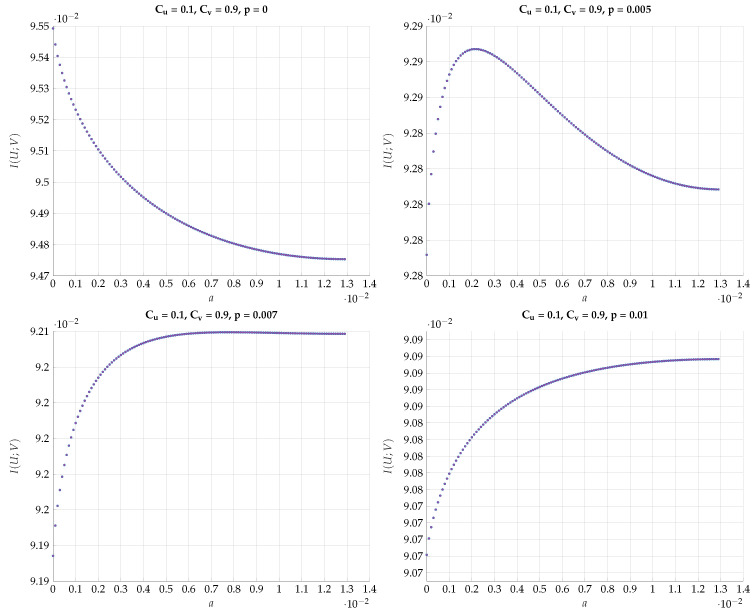
Maximal I(U;V) for fixed values $\left(C_{u},C_{v}\right)=\left(0.1,0.9\right)$ and different values of *p*.

**Figure 20 entropy-24-01321-f020:**
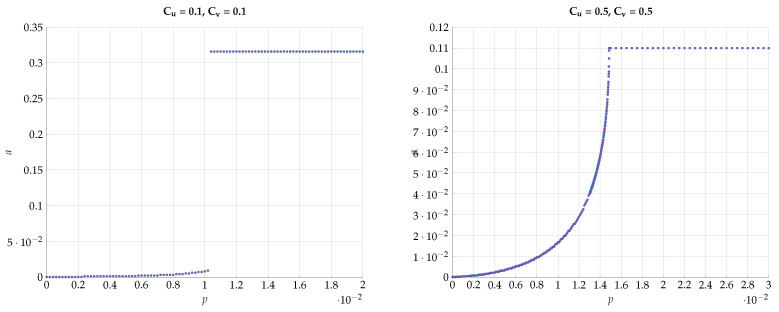
Optimal value of *a* for various values of $C_{u}$ and $C_{v}$.

**Figure 21 entropy-24-01321-f021:**
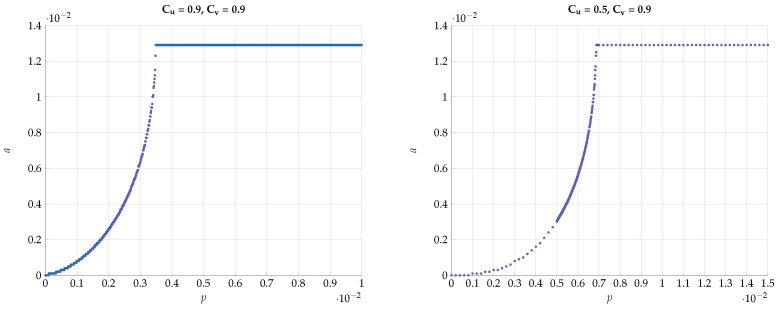
Optimal value of *a* for various values of $C_{u}$ and $C_{v}=0.9$.

**Figure 22 entropy-24-01321-f022:**
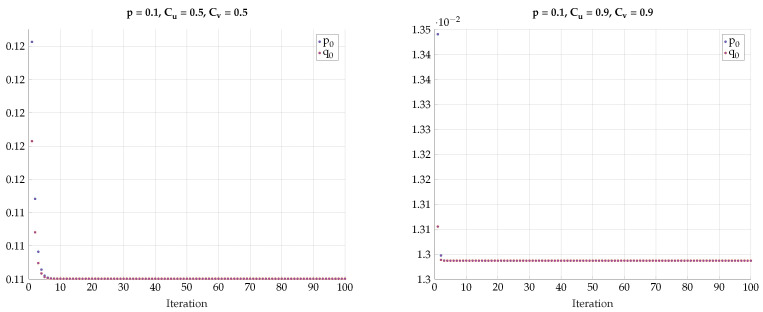
Alternating maximization with exhaustive search for various *p*, $C_{u}$, $C_{v}$.

**Figure 23 entropy-24-01321-f023:**
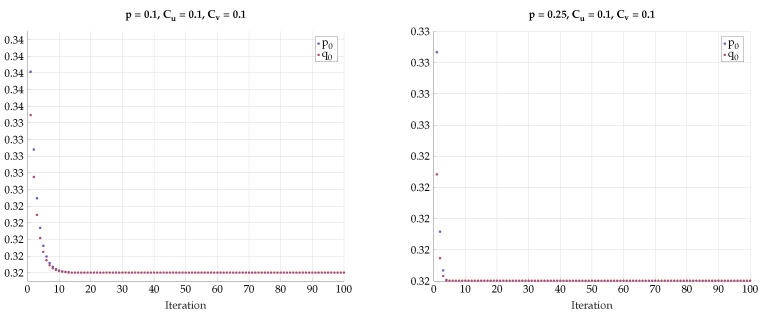
Alternating maximization with exhaustive search for various *p*, $C_{u}$, $C_{v}$.

**Figure 24 entropy-24-01321-f024:**
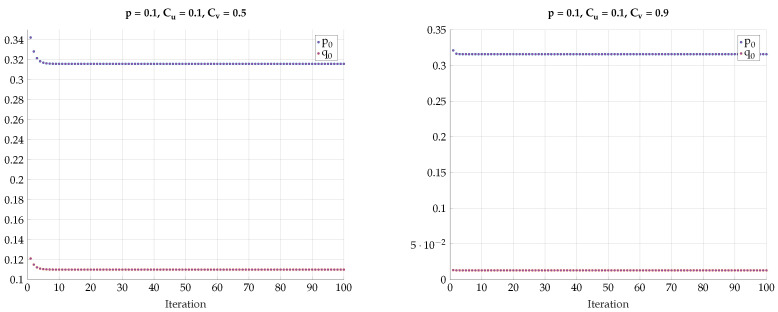
Alternating maximization with exhaustive search for various *p*, $C_{u}$, $C_{v}$.

**Figure 25 entropy-24-01321-f025:**
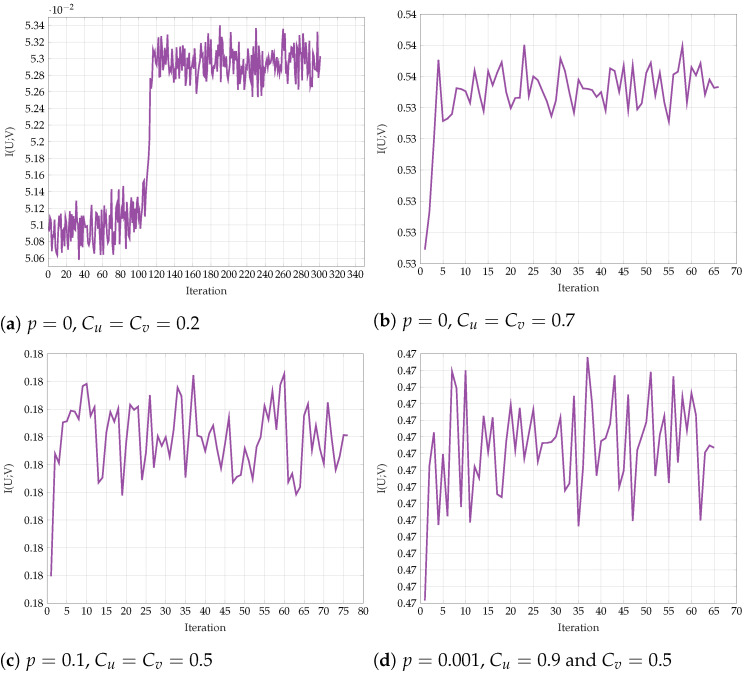
Convergence of I(U;V) for various values of *p*, $C_{u}$ and $C_{v}$.

**Figure 26 entropy-24-01321-f026:**
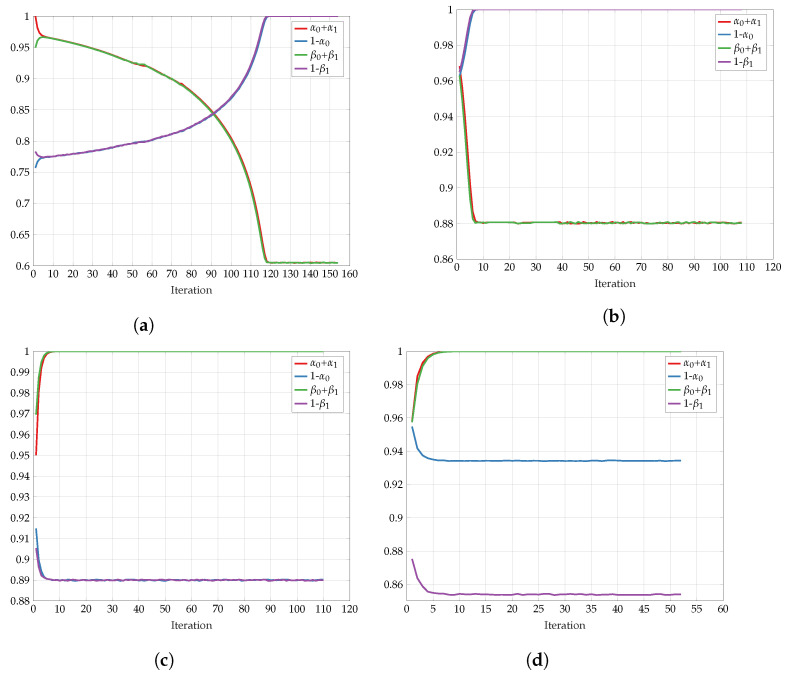
Convergence of I(U;V)
*p* with: (**a**) $C_{u}=C_{v}=0.2$, p=0; (**b**) $C_{u}=C_{v}=0.7$, p=0; (**c**) $C_{u}=C_{v}=0.5$, p=0.1; (**d**) $C_{u}=0.65,C_{v}=0.4$, p=0.1.

**Figure 27 entropy-24-01321-f027:**
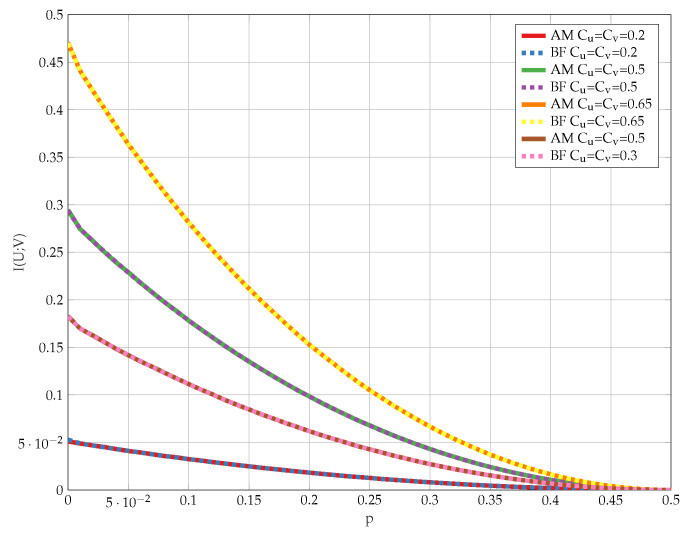
Comparison of the proposed alternating maximization algorithm and the brute-force search method for various problem parameters.

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
