# Peer review of "The Double-Sided Information Bottleneck Functionâ€"

_entropy, 2022, doi:10.3390/e24091321_

Round 1
Reviewer 1 Report
The manuscript is an extended version of the IEEE paper
"The Double-Sided Information-Bottleneck Function", 2021 IEEE International Symposium on Information Theory (ISIT), DOI: 10.1109/ISIT45174.2021.9517899
Astonishingly, the conference paper is not referenced in the manuscript. Compared to the conference paper, there is a significant added value of this journal version by providing mathematical proofs of theorems and lemmas as well as new numerical results. However, multiple paragraphs of the conference paper are re-used in a word-by-word fashion. I recommend to include the ISIT paper into the list of references, to mention the additional contributions of the journal paper and to appropriately reformulate the corresponding paragraphs.
Further comments:
- The definition in (2) seems to be not used in the manuscript and is obsolete.
- (11) contains a typo: the maximization must be performed w.r.t. T_{U | X}, not T_{Y | X}.
- first sentence in Sec. 5 (lines 350-353) remains unclear. Why is this conclusion meaningful? Please explain in more detail.
- The scaling/labeling of the y axes in Figs. 19, 20, 23, 24 shall be improved. The observed effects occur in a very limited range. It would be helpful for the reader if the authors provide additional information about the numerical stability of the algorithms and the reconstructability of the presented results.
- sentence below (36): is this still exhaustive search or alternating maximization (see caption of Figs. 23,24,25)?
- Fig. 26: provide values of p for subplots. A convergence in 3 subplots is hardly visible due to the scaling/labeling of the y-axis
Author Response
We would like to thank Reviewer 1 for his comprehensive and thorough review.
We have added the reference to our ISIT paper and reformulated the structure of the paper.
We further revised the final version of the paper based on his additional remarks.
Reviewer 2 Report
The paper presents an extension of the information bottleneck problem to having two sources of side information. The technical details appear to be correct. My comments were the following:
1. I might have missed this somewhere, but does the solution proposed by the authors reduce to the Tishby et al. method in the case of one source?
2. Line 37: please provide a reference for `biclusters'.
Author Response
We would like to thank Reviewer 2 for his valuable remarks. As follows our answers to his comments:
1. In both case, Tishby's and ours double-sided extension, there is a bivariate source (X,Y). The main difference is that in our setting there is an additional encoder connected to X that creates a compressed version U subject to an additional bottleneck constraint Cu. The metric we would like to optimize is the mutual information between U and V. We will add this clarification to the final version of the paper.
2. We have added a reference to the final version of the paper.
Reviewer 3 Report
This paper considers the problem of double-sided information bottleneck which is an extension of the standard information bottleneck problem into a two terminal setting.The work provides bounds on the double-sided information bottleneck and investigates various special cases, mainly the for jointly Gaussian and binary sources. The key analytical conclusions give the optimal test channels for both of these sources in the extreme case of high SNR (Gaussian) and low correlation (binary). The results show that these channels are basic symmetric channels (Gaussian and BSC) and demonstrate that these symmetric channels are not optimal in the other extreme case, i.e., low-SNR with Gaussian sources and high correlation with binary. The authors further give a conjecture for the optimal test channels in the high correlation case for binary sources. The conjecture indicates that the optimal test channels in this case are interestingly the asymmetricized version of the BSC, i.e., Z and S channels.
From technical viewpoint, the work is solid. The derivations (as much as I could check) are clear and correct. The results are also consistent with the intuition and have been presented very well. From presentation respect, the paper gives a very good literature review, a deep illustration of the earlier work, the related lines of research and connections of the topic to various applications. The writing is also very well. There are only few minor typos and writing errors which need to be fixed before publication, e.g., in Conjecture 3 the dot "." after \theta(C_u,C_v) should be removed, and in citation [6] "the The" should be "the". I would also find it more self containing, if the binary entropy function h_b(p) is directly defined in the notation, rather than in the main body of the paper.
Given the considerable contributions and the very good representation of the work, I would find the paper publishable in its current form, after fixing a few minor writing mistakes.
Author Response
We would like to thank reviewer 3 for his valuable comments. All his remarks are incorporated in the final version of the paper.